# Current Insights into m^6^A RNA Methylation and Its Emerging Role in Plant Circadian Clock

**DOI:** 10.3390/plants12030624

**Published:** 2023-01-31

**Authors:** Nguyen Nguyen Chuong, Phan Phuong Thao Doan, Lanshuo Wang, Jin Hee Kim, Jeongsik Kim

**Affiliations:** 1Interdisciplinary Graduate Program in Advanced Convergence Technology & Science, Jeju National University, Jeju 690756, Republic of Korea; 2Subtropical Horticulture Research Institute, Jeju National University, Jeju 690756, Republic of Korea; 3Faculty of Science Education, Jeju National University, Jeju 690756, Republic of Korea

**Keywords:** MTA, MTB, FIONA1, m^6^A methylation, m^6^A writer, m^6^A eraser, m^6^A reader, epitranscriptome, circadian clock

## Abstract

*N6*-adenosine methylation (m^6^A) is a prevalent form of RNA modification found in the expressed transcripts of many eukaryotic organisms. Moreover, m^6^A methylation is a dynamic and reversible process that requires the functioning of various proteins and their complexes that are evolutionarily conserved between species and include methylases, demethylases, and m^6^A-binding proteins. Over the past decade, the m^6^A methylation process in plants has been extensively studied and the understanding thereof has drastically increased, although the regulatory function of some components relies on information derived from animal systems. Notably, m^6^A has been found to be involved in a variety of factors in RNA processing, such as RNA stability, alternative polyadenylation, and miRNA regulation. The circadian clock in plants is a molecular timekeeping system that regulates the daily and rhythmic activity of many cellular and physiological processes in response to environmental changes such as the day-night cycle. The circadian clock regulates the rhythmic expression of genes through post-transcriptional regulation of mRNA. Recently, m^6^A methylation has emerged as an additional layer of post-transcriptional regulation that is necessary for the proper functioning of the plant circadian clock. In this review, we have compiled and summarized recent insights into the molecular mechanisms behind m^6^A modification and its various roles in the regulation of RNA. We discuss the potential role of m^6^A modification in regulating the plant circadian clock and outline potential future directions for the study of mRNA methylation in plants. A deeper understanding of the mechanism of m^6^A RNA regulation and its role in plant circadian clocks will contribute to a greater understanding of the plant circadian clock.

## 1. Introduction

Plants, as multicellular organisms, rely on a range of complex molecular mechanisms to control gene expression for normal development and stimuli response. Gene expression is controlled on two levels, which involve the amount of mRNA that is produced and the regulation of mRNA translation into proteins [1]. Recent studies have highlighted the importance of RNA modification as a crucial mechanism that dynamically modulates the cellular transcriptomic profile [2]. Currently, more than 200 types of RNA modifications have been identified [3], which are involved in various plant cellular and biological processes, including embryo development, shoot stem cell fate, floral transition, trichome morphogenesis, leaf initiation, and root development [4,5,6,7,8,9,10,11,12].

In eukaryotic mRNA, m^6^A is the most prevalent type of modification [13,14], which occurs through the addition of a methyl group (CH_3_) to the *N6* position of adenosines of the mRNA [15]. The presence of m^6^A in plants was first identified in maize and later in many other plant species such as Arabidopsis, wheat, and oats [16]. The occurrence of m^6^A in Arabidopsis mRNA was determined to vary between 0.45–0.65% of the total adenosine bases, with an estimated 0.5–0.7 m^6^A peaks per 1000 nucleotides [17]. A recent study reported that m^6^A sites can be detected in 32–35% of all expressed transcripts [13]. The frequency of m^6^A is unevenly distributed within the mRNA and is predominantly clustered near the stop codon and 3′ untranslated regions (UTRs) [18,19,20]. The m^6^A of plants is also found to be enriched near the start codon [17,21]. Many chloroplast-associated and photosynthesis-related genes showed an abundance of m^6^A sites located around the start codon, which suggests a unique function of m^6^A in plant photosynthesis [17,22].

Though these modifications have been detected since the 1970s, studies on m^6^A were initially limited due to the lack of methods for identifying m^6^A sites. However, with the discovery of the first m^6^A demethylase (fat mass and obesity-associated protein [FTO]) in 2011 [23], m^6^A has been determined to be a dynamic and reversible process that may have regulatory functions. Since then, several approaches have been developed to facilitate the functional study of m^6^A methylation. Based on the strategies to identify or recognize m^6^A sites, these approaches can be broadly divided into two categories: antibody-dependent methods such as methylated RNA immunoprecipitation (MeRIP)-seq [18], UV cross-linking and immunoprecipitation (miCLIP)-seq [24], or super-low-input m^6^A (SLIM)-seq [25] and antibody-independent methods such as deamination adjacent to RNA modification targets (DART)-seq [26], m^6^A-selective chemical labeling (m^6^A-SEAL)-seq [27], m^6^A-selective allyl chemical labeling (m^6^A-SAC)-seq [28], or nanopore-based direct RNA (DR)-seq [29]. Each of these methods has its own advantages and limitations, which have been discussed in detail in previous reviews [30,31].

As more efficient mapping methods are developed and more m^6^A-related genes are identified, the important roles that m^6^A plays in various biological processes are being gradually discovered. Although most m^6^A functional studies were conducted in an animal system, several reports have shown that m^6^A modification plays a crucial role in the regulation of plant development [4,32,33,34,35] and stress resistance [7,32,36,37]. In this review, we provide the current progress on the understanding of m^6^A biogenesis and function and its involvement in the regulation of plant circadian clock.

## 2. m^6^A in Plants

### 2.1. General Mechanisms

Generally, the m^6^A modification process involves numerous components, which include methylases, demethylases, and m^6^A-recognizing proteins (Figure 1). These are commonly termed m^6^A writers, erasers, and readers, respectively [15,38]. In plants and other eukaryotes, *N6*-adenosine methylation occurs through the binding of m^6^A writers to a highly conserved consensus sequence, RRACH (R = G/A; H: U/A/C) [10], which leads to the transfer of a methyl group from an S-Adenosyl Methionine (SAM) molecule onto the sixth nitrogen of the adenosine base [39]. However, the m^6^A abundance is significantly lower than that of the RRACH motifs, indicating that not all of these consensus motifs are associated with m^6^A modification and that the mechanism by which these motifs are selected for m^6^A modification remains elusive [40]. The reverse process of oxidatively removing the methyl group from the adenosine base is modulated by m^6^A erasers [39]. Finally, m^6^A-modified RNA is recognized by the m^6^A readers which subsequently utilize different complexes to regulate the RNA fate [41]. The Arabidopsis genome contains 29 m^6^A regulatory genes (6 writers, 6 erasers, and 17 readers) which encode for 55 proteins (8 writers, 14 erasers, and 33 readers) [42]. Next, we discuss in detail the structural features and the mechanistic insight of each m^6^A component.

### 2.2. m^6^A Writers

Generally, the methylase-containing protein complexes that are responsible for the *N6*-adenosine methylation of mRNA are commonly called m^6^A writer complexes [43]. Components of these complexes are evolutionarily conserved between the main eukaryotic lineages [4]. Studies on mammalian and *Drosophila* systems indicate that the m^6^A methylase complex consists of two sub-complexes, which are the m^6^A-METTL complex (MAC) and the m^6^A-METTL associated complex (MACOM) [44]. METTL3 and METTL14 together form the stable heterodimer that is MAC, while the MACOM consists of WTAP, RBM15, VIRMA, ZC3H13, and HAKAI [45]. METTL3, the first discovered methylase, harbors a catalytic subunit that serves as the main catalytic core for the methylation reaction [46,47]. Though METTL14 shares a high homology with METTL3 it is unable to bind with SAM but rather functions as a facilitator for the interaction between METTL3 and the RNA target [48,49]. MACOM interacts weakly with MAC to support the MAC functions, with each of its components performing a distinct function. WTAP binds directly to METTL3 to improve the activity of methyl transferase by ensuring the proper localization of the METTL3-METTL14 heterodimer [50]. RBM15 functions by recruiting the complex to the specific sites on RNA to promote methylation [51]. VIRMA mediates the preferential m^6^A mRNA methylation in the 3′ UTR and near-stop codons [52]. ZC3H13 is required for the nuclear localization of the complex [53], while HAKAI appeared to stabilize the WTAP and VIRMA protein [54].

In Arabidopsis, several orthologues of animal m^6^A writer components have been identified with similar functions and shown to interact with each other [8,38]. These orthologues include the mRNA adenosine methylase A (MTA; an orthologue of the methyl transferase 3 [METTL3]) [55] and B (MTB; an orthologue of METTL14) [8], the splicing factor FKBP12 interacting protein 37 kDa (FIP37; an orthologue of the Wilms tumor-1 associated protein [WTAP]) [8], the protein virilizer (VIR; an orthologue of the m^6^A methyltransferase-associated virus [VIRMA]/KIAA1429) [8], the putative ubiquitin E3 ligase HAKAI [8], the RNA-binding protein FPA (an orthologue of the RNA-binding protein 15 [RBM15]) [56], and the HAKAI-interacting zinc finger protein HIZ2 (a possible orthologue of the CCCH type 13 zinc finger protein [ZC3H13]) [57] (Figure 1). Protein sequence analysis has identified the specific conserved domains that are present in each m^6^A writer, which are MT-A70, WTAP, VIR_N, HAKAI, and CCCH domains in MTA/MTB, FIP37, VIR, HAKAI, and HIZ2, respectively [42,57]. The MT-A70 domain is required for the binding of substrate SAM, which is essential for the formation of m^6^A in mRNA [46]. Similar to their animal orthologues, plant m^6^A writers are mainly localized in the nucleoplasm [8]. Interestingly, though the knockdown of RBM15 reduced global levels of m^6^A [51], the knockdown of FPA, the RBM15 orthologue in Arabidopsis did not [56]. This may indicate the distinct roles of RBM15/FPA in m^6^A methylation in animal and plant systems.

Apart from m^6^A methyl transferases in the MTA-MTB complex which are responsible for the majority of m^6^A sites in *Arabidopsis* [8,10,32,55], several methyl transferases have also been identified, including METTL16 [58], METTL5 [59], or the zinc finger CCHC type containing 4 (ZCCHC4) [60]. While there is currently no information regarding the orthologues of METTL5 and ZCCHC4 in plants, FIO1 has been identified as the orthologue of METTL16 in Arabidopsis [58,61]. However, FIO1 is distinct from METTL16 since FIO1 does not methylate Arabidopsis SAM synthetases or affect their transcript expression levels. Instead, FIO1 installs m^6^A into U6 small nuclear (sn)RNA, a small subset of poly(A)^+^ RNA, and several phenotype-related mRNAs, thereby regulating the mRNA stability and many developmental processes [62]. Another example of independent methyl transferase that is found in Arabidopsis is MTC (an orthologue of METTL4) [63]. The structural comparison revealed that Arabidopsis MTC homodimer shared similar features with the METTL3-METTL14 heterodimer, suggesting that MTC functions as the m^6^A writer [63]. MTC displayed *N6*−2′-O-dimethyladenosine (m^6^Am) in vivo and *N6*-methylation of 2′-O-methyladenosine (Am) within single-stranded RNA in vitro [63]. However, whether these methyl transferases act independently or cooperatively with other proteins for m^6^A methylation requires further investigation.

Though the m^6^A consensus sequences (RRACH) frequently appear in mRNA (once every ~57 nucleotides), only a few are methylated. Additionally, despite the abundance of RRACH sequences, m^6^A only occurs in specific transcripts. As previously mentioned, the regulatory mechanisms of this transcript- and site-specific selectivity remain unclear. Two models have been proposed for the recruitment of m^6^A writer complex to the specific transcript, which is mediated by (i) transcription factors or (ii) histone modifications. Similarly, there are also two proposed models for site-specific targeting of m^6^A, which are (i) the RNA-binding protein (RBP)-mediated and (ii) RNA polymerase II (Pol II)-mediated recruitment models. These models have been extensively discussed in a previous review [64]. However, it is noteworthy that these models only account for a small fraction of m^6^A-containing transcripts in the cell, suggesting that there are unknown mechanisms responsible for most of the mRNA *N6*-adenosine methylation.

The m^6^A modifications occur in response to various internal and external stimuli such as heat shock, DNA damage, or stress [65,66,67,68,69]. The m^6^A writer complex is suitable for regulation at different levels, such as changes in the abundance of individual components or post-translational modifications (PTMs). The catalytically active subunit METTL3, which is targeted by miRNAs [70] and SUMOylation [71], is an obvious candidate for this regulation. Expression and stability of other m^6^A writer complex components appear to be METTL3-dependent [72] and can be affected by PTMs [73]. Overall, much knowledge regarding the regulatory role of this methyl transferase activity and m^6^A landscape remains to be elucidated.

### 2.3. m^6^A Erasers

m^6^A modification is a reversible process, by which the methyl group is removed catalytically by the m^6^A erasers in a dynamic, rapid, signal-dependent manner [74]. The m^6^A demethylases belong to the ALBHK protein family (a homolog of the α-ketoglutarate dependent dioxygenase AlkB family) [23,75]. To date, FTO and ALBHK5 are the only two identified m^6^A erasers in animals, although several members of the AlkB family have been documented to function by reversing m^6^A methylation [76]. Unlike FTO, which has no identified homolog in the plant system, the Arabidopsis genome contains 13 ALBHK family members with different subcellular localizations [77]. Five of these proteins are identified to be homologs of ALBHK5 including ALKBH9A, ALKBH9B, ALKBH9C, ALKBH10A, and ALKBH10B [6]. Among these, ALKBH9B is suggested to have demethylase activities in vitro and in vivo [7]. However, *alkbh9b* and *alkbh9c* mutants did not display any changes in the m^6^A/A ratio [6]. Another homologous protein, ALKBH10B, was shown to demethylate m^6^A in vivo and in vitro and the *alkbh10b* mutant consistently has increased m^6^A level [6]. Although m^6^A erasers have previously been suggested to play a crucial role in m^6^A functioning, they now appear to be limited to specific tissues and conditions [64].

### 2.4. m^6^A Readers

Eukaryotes can perceive m^6^A marks via the m^6^A readers. These m^6^A-binding proteins play a major role in the mechanism by which m^6^A regulates RNA metabolism. The characterization of m^6^A readers revealed that these proteins can be categorized into three classes based on their mechanism of recognizing m^6^A-containing RNA, which are (i) direct binding to the m^6^A base using a YT521-B homology (YTH) domain [78,79]; (ii) binding of the exposed single-stranded RNA motifs generated by m^6^A-induced structural changes [80,81]; or (iii) utilizing a common RNA-binding domain (RBD) and its flanking regions to recognize the m^6^A-modified transcripts [82]. Class I m^6^A readers are proteins that contain the YTH domain, which recognizes and binds to the *N6*-methyl group of adenosine via a hydrophobic binding pocket that contains conserved aromatic side chains [83]. The binding of class II readers to m^6^A-containing transcripts occurs via an m^6^A switch mechanism, in which the methylation destabilizes the Watson-Crick base-pairing and increases the accessibility of a single-stranded RNA-binding motif that is recognized by this class [80,81]. Class III readers lack of YTH domain and bind to m^6^A-containing transcripts in an RNA-structure-independent manner [82]. Though these are shown to utilize a common RBD and its flanking region for the recognition of m^6^A modifications, the exact mechanism behind these selective binding remains unclear. The different m^6^A-binding mechanisms indicate a broad and complex network of m^6^A-dependent pathways, and each class of m^6^A reader can be employed for specific targets or conditions.

Currently, there is no information regarding the homologs of class II and III readers in the plant system. In contrast, 13 proteins containing the YTH RNA-binding domain have been identified in the Arabidopsis genome [84]. These proteins can be categorized into two subgroups: (i) the evolutionarily conserved C-terminal region 1–11 proteins (ECT1–11) that belong to the YTHDF family [4] and (ii) the other two members that belong to the YTHDC family (ECT12 and the cleavage and polyadenylation specificity factor 30 [CPSF30]) [9,85]. Structural analysis revealed that the YTH domain of the first group is located near the C-terminus, in contrast to the other group whose YTH domain is located in the internal regions [86]. The conserved consensus sequence RRm^6^ACH is recognized by the YTH domain of ECTs. Concurrently, the intrinsic disorder region within the ECTs forms a stable interaction with the U-rich sequence in adjacent regions, thereby controlling the occupancy of the binding adjacent to m^6^A with different RNA-binding factors. These ECT-binding targets are mainly involved in the translation and metabolic processes [4]. Arabidopsis CPSF30 forms two splice isoforms via alternative polyadenylation: (i) a ~28 kDa isoform (CPSF30-S) that harbors three zinc finger domains and is homologous to mammalian CPSF30 and (ii) a ~70 kDa isoform (CPSF30-L) that has an additional YTH domain that is unique to plants [87,88]. The CPSF30 binding site is predominantly located in the mRNA 3′ UTR region and its m^6^A binding ability enhances the formation of CPSF30 liquid-like nuclear bodies to regulate polyadenylation [85].

### 2.5. m^6^A Regulates RNA Activity

m^6^A employs diverse mechanisms to regulate RNA metabolism, which involves different aspects such as stability, structure, translatability, localization, or splicing (Figure 1). However, most m^6^A studies regarding its molecular functions were conducted in animals. The impact of m^6^A on plant RNA activity is limited. Current evidence suggests that m^6^A in plants, like in the animal model system, also mediates the stability of RNA under various conditions. For example, the knockdown of m^6^A components such as the writer MTA [6,32] or reader ECT2 [11] and CPSF30 [85] in Arabidopsis was shown to accelerate the degradation of target transcripts. Similarly, the writer FIP37 destabilized mRNA of the key shoot meristem genes [10] while demethylation by the ALKBH10B eraser promoted the stability of its targets [6].

Several underlying mechanisms have been proposed to explain the m^6^A roles in RNA stability. The m^6^A binding by the mammalian reader YTHDF2 may directly or indirectly regulate transcript stability by either recruiting the CCR4-NOT deadenylase complex to destabilize the m^6^A-containing mRNA [89] or promoting the translocation of mRNA from the translation machinery to processing bodies for degradation [90]. In another study, m^6^A was found to regulate the interaction between mRNA and the mRNA stabilizer human antigen R (HuR) in a distance-dependent manner [91]. In close proximity, m^6^A promotes HuR binding and conversely decreases HuR binding when far apart [91]. The regulatory roles of m^6^A on mRNA stability may also lead to structural changes in mRNA, under certain conditions. For example, m^6^A increases the stability of mRNA and subsequently leads to a decrease in structural complexity in response to a change in salinity [92]. m^6^A may impact RNA structure by promoting the transition from paired to unpaired RNA, as suggested by the less structured mRNAs with the m^6^A consensus motif GGACU in wild-type mouse embryonic stem cells (mESCs) compared with that of *METTL3*-knockout mESCs [93].

Another potential mechanism that m^6^A may utilize to regulate RNA metabolism is alternative polyadenylation (APA), which has been suggested in several studies. Reducing expression of *VIR*, an m^6^A writer in Arabidopsis negatively affected mRNA 3′s end formation and subsequently resulted in the preferential proximal poly(A) site selection [13]. The reader CPSF30 was also found to regulate APA and poly(A) site selection [85,94]. Additionally, m^6^A is necessary to maintain transcriptome integrity by guiding site-specific mRNA polyadenylation in target genes with intrinsic transcription termination and polyadenylation defects [95]. This APA pathway termed the m^6^A-assisted polyadenylation requires the writer FIP37 and reader CPSF30-L for proper functioning [95]. Studies on other plant species support the involvement of m^6^A in the APA pathway, as m^6^A modification can affect poly(A) site selection in maize [96].

There are lines of evidence that suggest the interaction between m^6^A and other regulatory mechanisms of mRNA. m^6^A modifications were found to affect the primary-microRNA (pri-miRNA) secondary structure and recruitment of microprocessor complexes to the pri-miRNA in Arabidopsis [97]. Furthermore, the m^6^A writer MTA also interacted with proteins involved in miRNA biogenesis such as RNA Polymerase II and TOUGH [97]. This observation is consistent with the animal system, as *N6*-adenosine methylation of pri-miRNA mediated by METTL3 facilitated the DGCR8 microprocessor complex pri-miRNA processing [98]. Another m^6^A component involved in this process was the nuclear m^6^A reader hnRNPA2B1 [99]. In the opposite direction, miRNA machinery plays a role in regulating m^6^A formation, possibly by modulating METTL3-mRNA binding [100]. Crosstalk between histone modification and m^6^A has also been proposed in a previous study. H3K36me2 and m^6^A were found to co-occur near the 3 UTR region and affected the global level of each other in Arabidopsis [101]. Furthermore, the direct interaction between an H3K36me2 writer SET DOMAIN GROUP 8 (SDG8), and an m^6^A writer protein FIP37 was observed. Still, further investigation is required to fully understand this crosstalk. In conclusion, though has been indicated in many previous studies, the underlying mechanisms for the regulatory roles of m^6^A in RNA metabolism require further investigation, especially in plant systems.

## 3. m^6^A in the Plant Circadian Rhythm

### 3.1. Plant Circadian Clock and m^6^A Methylation

Plants, as sessile organisms, evolve to sense and predict environmental changes. The most dramatic changes in the environment are day-night cycles due to the earth’s rotation and many processes in plants are synchronized with their circadian cycles. The circadian clock is the endogenous timekeeper that governs and coordinates these rhythmic processes to increase plant fitness. This is based on the observation that altered period length or out-of-phase conditions in plants interfere with their optimal growth or defense against abiotic and biotic stress [102,103]. The fundamental and molecular aspects of the circadian clock rely on the transcriptional feedback among the central oscillator components with morning components, CIRCADIAN CLOCK-ASSOCIATED1 (CCA1)/LATE ELONGATED HYPOCOTYL (LHY), and an evening component, TIMING OF CAB EXPRESSION1 (TOC1) (Figure 2) [104]. Additional transcriptional feedback loops involve morning and evening complexes that link the central clock components. The morning components, such as the PSEUDORESPONSE REGULATOR9 (PRR9) and PRR7, act to reset the clock in the morning by inhibiting the transcription of *LHY* and *CCA1* [105]. The evening complex with the LUX ARRHYTHMO (LUX), EARLY FLOWERING3 (ELF3), and ELF4 act to maintain the clock in the evening by repressing the transcription of *PRR9* and *PRR7* [106]. Cycling activators such as NIGHT LIGHT-INDUCIBLE AND CLOCK-REGULATED GENE1 (LNK1)/LNK2 and REVEILLE8 (REV8) are involved in the activation of evening complex components including *LUX*, *ELF3*, and *ELF4* [107,108]. Post-translational feedback regulation has a role in the maintenance of the circadian clock. For example, ZEITLUPE (ZTL) promotes the degradation of TOC1, while the GIGANTEA (GI) and PSEUDORESPONSE REGULATOR3 (PRR3) function to stabilize and protect TOC1 [109,110]. Overall, this complex network of transcriptional and post-translational feedback loops allows the clock to maintain a consistent and robust circadian rhythm in Arabidopsis.

In plants, the circadian clock controls the rhythmic expression of thousands of genes through transcriptional activation or suppression by multiple transcription factors [112]. Recent studies using RNA-seq analysis have highlighted the importance of post-transcriptional regulation of mRNA, including alternative splicing, polyadenylation, mRNA nuclear export, mRNA degradation, and mRNA methylation, in the rhythmic expression of these genes [113,114,115]. In particular, m^6^A methylation has been proposed as a new and emerging layer of dynamic mRNA regulation in the circadian clock. In mammalian cells, many well-known clock genes, including Clock, Periods (*Pers*), Albumin D-site-binding protein, Nuclear receptor subfamily 1 group D member 1 (*Nr1d1*), and Casein kinase 1 delta (*CK1δ*), contain multiple m^6^A methylation sites on their transcripts [116,117]. Silencing of the m^6^A methylase METTL3 expression lengthened the circadian period through a delay in the nuclear export of mRNAs of the clock genes *Per2* and Aryl hydrocarbon receptor nuclear translocator-like (*Arntl*) [116]. Cryptochrome 1 (CRY1) and CRY2 are core components of the mammalian circadian clock and interact with an m^6^A eraser FTO, which is required to maintain global m^6^A methylation levels [118]. In addition, *cry1cry2* knockdown mice with a complete loss of circadian rhythm have significantly lower levels of m^6^A and a loss of the circadian rhythm of m^6^A [118]. In plants, daily oscillation of mRNA transcripts of the writers and erasers and accumulation of global m^6^A levels at midnight have been observed in seagrasses *Cymodocea nodosa* and *Zostera marina* [119].

Recent advancements in molecular technologies for direct detection of RNA methylation, such as nanopore mRNA-seq, miCLIP-seq, and m^6^A-seq, have enabled high-throughput quantitative analysis of the methylation levels and accurate identification of deposition sites on mRNA in various developmental conditions [13,120]. m^6^A methylation is more frequently found in mRNAs of clock-controlled genes (CCGs) and many mRNAs of clock components are m^6^A methylated in Arabidopsis [17,120]. m^6^A is the most prominent and dynamic type of RNA methylation and determines the fate of CCG mRNA, including the mRNA stability, splice-site choice, and RNA 3′ end formation [13,120]. The importance of m^6^A in the regulation of the circadian clock in plants has been emphasized by several recent studies on the genome-wide analysis of the function of (i) CRY1/2-mediated MTA-MTB-FIP37 methylase complexes [112,120]; (ii) VIR, another conserved m^6^A writer complex component [13]; and (iii) FIO1, a homolog of the m^6^A writer METTL16, in the regulation of the circadian clock and related physiology (Figure 2) [62,121,122,123,124].

### 3.2. Circadian Clock Regulation through m^6^A Methylation by a General m^6^A Writer Complex

Deposition of m^6^A on the majority of mRNAs is catalyzed by a multiprotein RNA methylases complex, which consists of two methylases (MAC) including MTA and MTB, as well as the additional proteins (MACOM) including FIP37, VIR, and HAKAI [8,74]. MTA, MTB, and FIP37 are the core components of the m^6^A methylase complex as counterpart subunits of METTL3, METTL14, and WTAP in the mammalian METTL3/14-type general m^6^A writer complex [38].

In Arabidopsis, the m^6^A deposition is more commonly detected in the mRNAs of CCGs, particularly those involved in chloroplast-related or photosynthesis processes, with 57% of CCG mRNAs relative to the 41% of expressed mRNAs [120]. m^6^A methylation is also more frequently found in mRNAs of clock regulatory genes, with a 1.4-fold enrichment. These include the morning components, *CCA1* and *LHY*, and the evening components, *GI* and *ELF3* [120]. To understand the role of the general m^6^A writer complex in the circadian clock, the m^6^A profiles and circadian rhythms should be examined in these mutants. However, these loss-of-function mutations result in embryonic lethality, which is a major obstacle in exploring the role of m^6^A in the circadian clock. Instead, partially rescued plants expressing the embryo-specific *ABSCISIC ACID INSENSITIVE3* (*ABI3*) promoter-driven MTA in the *mta* mutant (*ABI3::MTA/mta*) were used to reveal the role of the MTA methylase in the circadian clock [120,125]. These plants demonstrated a 90% reduction in m^6^A methylation in leaves and a long-period rhythm in white light with reduced m^6^A methylation in the 3′ UTR of *CCA1* mRNA. Interestingly, MTA and FIP37 physically interact with CRY1 and CRY2, blue light receptors, and MTB also interacts with CRY2 [55,112,120].

CRY1 and CRY2 mediate blue light input in the circadian clock depending on the fluence rate, with CRY1 for strong light and CRY2 for weak light [126,127]. However, the detailed mechanisms of how CRY1/2 blue light inputs are transduced to the circadian oscillators are not clearly understood and may include GI and PIFs [128,129]. Recently, it has been reported that CRY2 interacts with MTA to elicit m^6^A methylation of the CCGs to mediate the blue light input to the oscillator [120]. CRY-mediated blue light induces m^6^A modification in 10% mRNAs within a few hours in a fluence rate-dependent manner [112,120]. In blue light, CRY1 interacts with FIP37 and accelerates the m^6^A-mediated mRNA degradation of *PIF3*, *PIF4*, and *PIF5* which are negative regulators of light and circadian clock-mediated hypocotyl growth [112]. On the other hand, CRY2 forms acute and dynamic blue light-mediated photobodies that concentrate molecules in the complex through liquid-liquid phase separation [120]. CRY2 recruits MTA, MTB, and FIP37 in the photobodies within a few seconds, which induces the mRNA m^6^A modification of ten core clock component genes including *CCA1* and enhances their mRNA stability. The CRY2-mediated photobodies promote the enrichment of m^6^A writers and are regarded as another photo-regulatory layer in the regulation of the circadian clock [120,130].

Although CRY2 and the MTA-MTB-FIP37 methylase complex form photobodies and transfer the blue light input to the circadian clock, detailed analysis of their circadian clock phenotypes indicates that an additional and separate role of MTA in regulating the circadian clock exists [120]. The *cry1cry2* double mutant showed reduced responses to period shortening effects by increased intensity of blue light, indicating the involvement of CRY1/2 in parametric entrainment of blue light input to the circadian clock through continuous modulation of the circadian clock by recognizing the intensity of blue light. However, *mta* showed a similar period shortening responsiveness as the wild type in blue light, and a longer period in white light than in blue light, indicating that this additional role may be mediating other light input or the modulation of the core clock oscillator. However, the effect of blue light-induced photobodies on the dynamic mRNA m^6^A modification of circadian regulatory genes remains unknown.

VIR, another conserved component of the general m^6^A writer complex, is also linked to circadian clock regulation [13]. The *vir-1* mutant, a weak mutant allele of the *VIR* gene, exhibited a remarkable reduction in the m^6^A ratio to between 5–15 % in 3′ UTR and further exhibited multiple developmental defects as well as a lengthened circadian period. In *vir-1*, the mRNA abundance in the circadian core clock components including *PRR7*, *LNK1/2*, and *GI*, which had m^6^A modification, was increased. Unlike with the *mta* mutant, the *CCA1* level was increased in *vir-1*, which is associated with a long-period rhythm in *vir-1*. VIR was also found in splicing speckles, where it was co-localized with the splicing factor SR34, but the alteration in splicing patterns of mRNA in the *vir-1* mutant was minor [8,13]. In contrast, *vir-1* mutation led to a global shift of the poly (A) tail length distribution of CCGs [13]. Additionally, *vir-1* mutation abolished the 3′ end formation in mRNAs, which is similar to the function of human VIRMA, an orthologue of VIR, in the polyadenylation selection [52]. Human VIRMA associates with the polyadenylation cleavage factors, CPSF5 and CPSF6, which facilitates the selection of proximal polyadenylation sites within the 3′ UTR of mRNAs. Plant VIR might regulate the stability of mRNA encoding circadian clock genes through nonsense-mediated decay of mRNA with lengthened 3′ UTRs or aberrant poly(A) tail lengths.

Although MTA and VIR function in the same methylase complexes and *mta* and *vir-1* mutations have similar lengthening effects on the circadian period, their functional role in regulating the circadian clock appears to be different [13,120]. MTA and VIR mediate the stabilization and 3′ end formation of their target mRNAs, respectively, despite their similar preference for m^6^A modification at the 3′ UTR. Interestingly, the levels of CCA1 mRNA, which are the result of these mutations, are opposite in their mutants relative to the wild type. This may be due to the cumulative effect of the altered activity of circadian clock genes, which are selectively guided by MTA and VIR. A more detailed investigation into the molecular effects of these mutations on clock-regulatory genes may reveal how they cooperate and act independently in regulating the circadian clock. It has been observed that m^6^A methylation of mRNAs can be regulated by the circadian clock, as global methylation profiles peak at midnight in marine plants [119]. Interestingly, the expression of several components in the general m^6^A writer complex, including *MTA*, *MTB*, and *HAKAI*, is diurnally regulated with a peak time at night and *MTB* expression is regulated by the circadian clock (Figure 3a,d) [131]. Further investigation into whether m^6^A methylation in CCGs is regulated by the circadian clock can suggest the dynamic role of m^6^A methylation by general m^6^A writer complexes in regulating the circadian clock.

Overall, it is evident that m^6^A methylation induced by the MTA-MTB methyl writer complex is essential for the photo-input or maintenance of the plant circadian clock, but the dynamic regulation and the role of m^6^A readers and erasers in regulating m^6^A methylation need to be established.

### 3.3. FIO1, a Core Clock Component as m^6^A Methylase

FIO1 was first identified by the screening of photoperiodic flowering responses and was further characterized as a core clock component for the maintenance of the proper circadian period [61]. FIO1 is predicted to be a tentative methylase with a SAM binding domain and an orthologous protein to mammalian METTL16. Mammalian METTL16, another type of m^6^A methylase, is known to function as a specific m^6^A writer on the mRNA MAT2A and the U6 snRNA without uncertainty, although a hundred candidate mRNAs could be predicted as its targets through a genome-wide m^6^A IP-seq [133]. MAT2A, the first identified target of METTL16, is the key enzyme in synthesizing SAM, which is required as a methyl donor for most cellular methylation events [134]. METTL16 establishes homeostasis of SAM through the stabilization of MAT2A through direct deposition of m^6^A on its mRNA when the availability of the SAM is limited in the cell. Another target of METTL16 is U6 snRNA which is an RNA component of U6 ribonucleoprotein in the spliceosome [135]. METTL16 binds and deposits a single m^6^A at the UACAGAGAA site within the stem-loop structure in the U6 snRNA, which is required for its incorporation into U4/U6 snRNA. METTL16 mediates the stability and splicing sites of the mRNAs by influencing the U6 spliceosome [136,137]. Recent research on FIO1 suggests that it possesses *bona fide* methylase activity with the catalytic motif with a stretch of amino acids NPPF for the binding of adenine substate RNAs in vitro and in vivo [62,123,124].

Recently, several reports have highlighted how FIO1 mediates m^6^A methylation by regulating various circadian clock-related developmental processes including photomorphogenesis and floral transition (Figure 2) [62,121,122,123,124]. In *fio1* mutants, about 10%–15% of m^6^A in the total mRNA and 10–60% of the U6 snRNAs were reduced in seedlings, rosette leaves, and floral buds, which were relatively minor effects compared to the 80%–90% m^6^A loss in *fip37* and *mta*. Global profiling of m^6^A methylation in *fio1* mutants revealed more than 2-fold hypomethylated m^6^A peaks in around 1000–2500 mRNAs, which are mainly located around the 3′ UTR and near the stop codon. In Arabidopsis, FIO1 installs an m^6^A at the consensus sequence of U6 m^6^A motif ACAGA and plant-specific mRNA m^6^A motifs GGACC and UGUAU, with higher activity in the consensus sequence and different preferential activity for single-stranded over-stem-loop structures in U6 snRNA [62,123]. It is noted that multiple motif variants have been found in FIO1-mediated methylation sites with YHAGA (Y = C/U; H = C/A/U) in the CDS region [121]. The GAACU and UGUAA consensus sequences in the 3′ UTR [123], and DRACH (D = A/G/U; H = A/C/U) in the 3′ UTR [124]. Interestingly, there were no global pattern changes in the remaining m^6^A between the wild type and *fio1* mutant, indicating that FIO1 has no preferential choice in m^6^A position and motif. FIO1 has a lesser effect on global m^6^A levels, but a higher m^6^A reduction in specific groups of mRNAs, indicating that FIO1 likely mediates certain biological and physiological processes [62,121,122,123,124]. Gene ontology analysis of the differentially expressed genes and genes with hypomethylated mRNA in *fio1* mutant revealed characteristics of circadian rhythm and flower development [62]. These results support that FIO1 regulates the m^6^A deposition in mRNAs of specific target genes involved in certain biological processes including the circadian rhythm.

The *fio1* mutant exhibits several developmental phenotypes, including a long circadian period, early flowering, and long hypocotyls, which may be due to an aberrant circadian rhythm or an independent function of FIO1 in developmental processes [61]. Based on the differential expression and hypomethylated mRNAs in the *fio1* mutant, many important regulatory genes involved in the circadian clock, floral transition, and photomorphogenesis are estimated to be the direct targets of FIO1 (Figure 2). For example, FIO1 affects the m^6^A modification of transcripts of *CCA1* and *LHY* in the circadian clock pathway, *CO* and *CRY2* in the photoperiodic flowering pathway, *FLC* in the autonomous and vernalization flowering pathway, *SOC1*, *SVP*, *SPL3,* and *SEP3* in the floral integration pathway, and *PIF4* in photomorphogenesis [62,121,123]. However, the effectiveness of m^6^A modification by FIO1 on the target mRNAs is different among them, with the degradation of *LHY*, *CO*, *CRY2*, *SOC1*, *SPL3*, *SEP3*, and *PIF4* mRNAs, and stabilization and altered splicing of *SVP* and *FLC* mRNAs [123]. Additionally, RNA-IP with FIO1 contributes to revealing a direct target of FIO1 as evidenced by the association of FIO1 with *SOC1*, *SVP*, *CCA1*, and *LHY* mRNAs [121], *PIF4*, *CRY2*, *CO*, and *FLC* mRNAs [62], and *SPL3* and *SEP3* mRNAs [123]. The effect of FIO1 on a wide range of targets supports the concept that FIO1 might be independently involved in various physiological processes through its m^6^A methylation of various target transcripts. However, several additional clock regulatory genes including *ZTL*, *LKP2*, *LIP1*, and *LCL1* [62], *ELF3*, *TIC*, and *PHYA* [123], and *WNK1, CKB3*, and *CRY1* [121] were predicted to be potential targets of FIO1, because their mRNAs were hypomethylated in *fio1* mutants, although their expression was not dramatically or consistently affected. Additionally, gene ontology analysis suggests that the regulation of the circadian clock is enriched in hypomethylated mRNAs in *fio1* mutants [62]. In this regard, it cannot be excluded that FIO1 plays a specific role in the regulation of the circadian clock through m^6^A methylation.

FIO1 is required for the maintenance of the period length in the core clock, but it is not clear how FIO1 contributes to the proper functioning of the circadian clock. As FIO1 functions as an m^6^A methylase, it may be involved in the depositing of m^6^A on the mRNAs of central oscillator genes, which affects their stability. *CCA1* and *LHY* mRNAs are representative targets of FIO1-mediated m^6^A methylation, but their mRNA levels are not consistently affected in *fio1* mutants in different studies, potentially due to different experimental conditions or alleles used. The cumulative effect of m^6^A on a subset of FIO1 target mRNAs induces changes in RNA stability, alternative 3′ ending, or splicing, which can modulate the activity of the circadian rhythm [113,114,115,138]. Alternatively, other effects of m^6^A methylation on target mRNAs such as changes in histone modification, nuclear retention, or translational activity, which have not been examined in these studies, may be involved [101,113,116,139].

Another important factor necessary to understand the role of FIO1 in regulating the circadian clock is whether the methylation activity of FIO1 is dynamically regulated by the circadian clock. FIO1 mRNA and protein levels are not controlled by the circadian clock [61]. It is possible that FIO1 generates a dynamic methylation profile on the mRNAs through its association with other m^6^A methylases, such as MTA-MTB m^6^A methylase complex and MTC m^6^A methylase, whose component and own expression, respectively, are regulated in a rhythmic manner (Figure 3a,d) [131]. Alternatively, FIO1 may have rhythmic collaborators that assist it to regulate the dynamic m^6^A methylation. Expression of genes encoding putative RNA methylases (AT5G51130 and AT5G10620) is diurnally rhythmic and their orthologous proteins can interact with METTL16 in humans and *Drosophila* [140,141,142]. The rhythmic features of m^6^A CCGs mRNA could be regulated in other layers of m^6^A methylation processing by m^6^A readers or erasers, which might act on different Zeitgeber times (Figure 3b,c,e,f). Global m^6^A methylation and expression profiles in *fio1* mutants under diurnal or free-running conditions can provide a comprehensive and accurate evaluation of the dynamic role of FIO1 on targets in the circadian clock.

FIO1 shares structural similarities with the mammalian METTL16, but their functional effect on m^6^A methylation might differ. This may be due to the difference in their subcellular localization; FIO1 is localized exclusively in the nucleus, while METTL16 protein exists in both the nuclear and cytoplasmic regions and also in the nucleolus in the G1/S phase [61,143,144]. The cytosolic mammalian METTL16 has additional functions in enhancing the translation efficiency of transcripts through interaction with the eukaryotic translation machinery beyond m^6^A deposition [133,144]. However, FIO1 is likely to function as a nuclear methylase in the nucleoplasm, although the possibility of residual cytosolic FIO1 having additional functions cannot be excluded. In contrast to METTL16, FIO1 did not induce changes in the global alternative splicing patterns, with only 43 alternative splicing events, although FIO1 also methylated m^6^A in U6 snRNAs [120]. Additionally, the m^6^A in four mRNAs that encode for the synthesis of SAM was not affected by FIO1 [62,124]. Importantly, FIO1 installs m^6^A in plant-specific m^6^A motifs beyond the consensus sequence motif of U6 snRNAs.

A recent report by Simpson and colleagues (2022), however, highlighted the importance of FIO1-mediated m^6^A modification in U6 snRNA rather than in mRNAs [124]. The m^6^A deposition on U6 snRNA by FIO1 is essential for accurate splicing through the preferential selection of the canonical site from the two major 5′ splice sites. RNA-seq analysis of the *fio1* mutant revealed widespread deregulation of pre-mRNA splicing by more than 2300 alternative 5′ splicing site selections, which is different from other reports [120,121]. Although m^6^A levels in more than 2800 sites are altered in the *fio1* mutant, over 85% of them overlapped with the hypomethylated sites in the *fip37* mutant, implying that FIO1-dependent methylation on mRNAs is relatively small. This might be due to the indirect effect of a 40% loss of MTB activity with premature termination due to altered splicing. Additionally, splicing changes in the mRNAs of several circadian clock genes, *LHY* and *WNK1,* and additional clock regulators, including *HOS1* and *SAR1,* were found in *fio1*. Interestingly, *hos1* and *sar1* mutants showed early flowering phenotypes with high levels of *CO* and reduced levels of *FLC* expression [145,146] and lengthened circadian periods [147], which is similar to that of the *fio1* mutant. In addition, global changes to the alternative splicing in loss-of-function mutants of the splicing of factor PRMT5, Type II protein arginine methyltransferase 5 [148,149], and SPLICEOSOMAL TIMEKEEPER LOCUS1 (STIPL1) [150] lengthened the circadian periods, which supports the indirect effect of the hypomethylation of U6 snRNA in *fio1* being responsible for its aberrant circadian rhythm. However, FIO1 can directly interact with their targets and affect the m^6^A levels in their mRNAs [62,121,123], supporting that FIO1 installs direct m^6^A deposition on specific mRNAs for the regulation of developmental processes. Regardless, FIO1 is not likely a major writer of m^6^A modifications but rather a more selective methylase that acts on specific mRNAs and U6 snRNAs for the regulation of the circadian clock. Further investigations into the biological implications of FIO1-mediated m^6^A on U6 snRNA and its target mRNAs are required.

## 4. Conclusions and Future Directions

In recent years, m^6^A has emerged as an essential mechanism for RNA metabolism regulation. However, a substantial amount of m^6^A studies have been conducted in animal systems, but there is a lack of information regarding the mechanistic insights of this RNA modification in plants. Although it is highly conserved among species, plant m^6^A machinery also contains distinct features or components compared to that of the animal systems such as the plant-specific CPSF30-L isoform [87,88] or the different roles of RBM15 and FPA in animal and plant systems [51,56]. Studies on the RNA regulatory mechanisms of plant m^6^A in conjunction with animal m^6^A may provide other perspectives on the conserved and distinct functions of m^6^A in plants and animals. Furthermore, the number of identified m^6^A proteins in plants is relatively low compared to that of animals, which is also an obstacle to fully understanding the functions of m^6^A in plants. Further investigation into the mechanisms for the site- and transcript-specific selection of m^6^A modification is an interesting direction to be focused on in the future.

The development of novel sequencing technologies has facilitated the functional studies of m^6^A and provided considerable insights into this regulatory mechanism. However, there are still limitations in these methods that hinder the m^6^A studies such as the large amounts of inputs, lack of stoichiometric information, and low specificity and efficiency. Furthermore, many of these approaches have not been tested on plant systems. Therefore, the development of an improved, plant-specific method may accelerate the progress in studying m^6^A in plant systems.

The plant circadian clock is a complex and dynamic system that is coordinately regulated at multiple levels. m^6^A methylation, a newly emerging layer of epitranscriptomic regulation, has been observed to be prevalent in transcripts of photoreceptors, clock regulatory genes, and CCGs, indicating its role in various hierarchical structures of the clock, including inputs, the central oscillator, and outputs. Genetic evidence from mutant studies for methyl writers has demonstrated the importance of m^6^A methylation in the regulation of the plant circadian clock.

There is still much to understand regarding the effects and mechanisms of m^6^A modification on transcripts in clock regulatory genes and how it influences the regulation of the plant circadian clock. Further research should delve deeper into this topic and explore the kinetics of RNA methylation and demethylation in relation to the circadian rhythm and specific regulatory pathways involved in the circadian clock. The function of m^6^A erasers or readers, as well as other m^6^A writers, in regulating the circadian clock and their related processes also needs to be clarified. Further investigation into the impact of m^6^A modification on RNA metabolism and key circadian RNA molecules will improve our understanding of the molecular mechanisms underlying the plant circadian clock.

## Figures and Tables

**Figure 1 plants-12-00624-f001:**
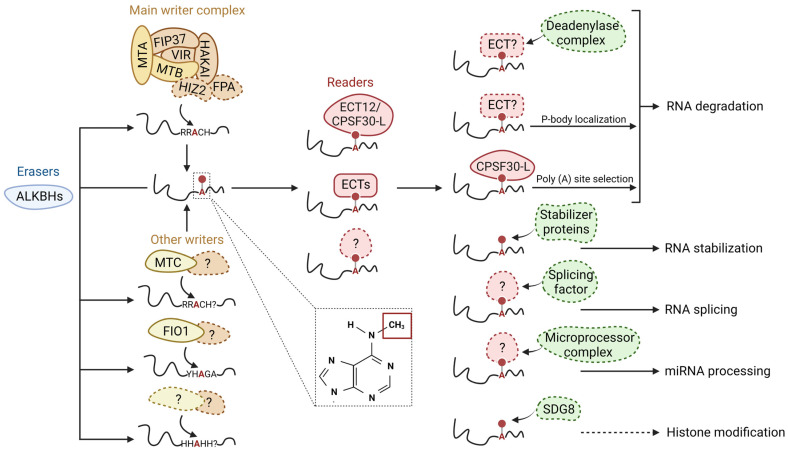
Schematic model of m^6^A modifications and its regulatory roles in RNA metabolism. m^6^A writers, erasers, readers, and other proteins/protein complexes are illustrated in yellow, blue, red, and green, respectively. The colorful circles with dashed lines indicate unidentified/unconfirmed components/factors/regulations in the plant system. m^6^A writers install a methyl group at the N-6 position of an adenosine base after binding to the RRACH (R = G/A; H: U/A/C) consensus sequence. Various plant writers have been identified along with their homologs in the animal system. The presence of other plant writers and motifs, should they exist, remains to be identified. m^6^A modifications can be removed enzymatically by erasers. Plant erasers contain only members from the ALBHKs family. m^6^A readers are responsible for recognizing m^6^A-containing transcripts. Plants readers include members from the ECT family and CPSF30. Structural changes in RNA after m^6^A modification or the recognition of m^6^A modifications by m^6^A readers can facilitate or inhibit the interaction between RNA and various protein complexes, thus directly or indirectly regulating RNA metabolisms including RNA degradation, RNA stabilization, RNA splicing, miRNA processing, and histone modification.

**Figure 2 plants-12-00624-f002:**
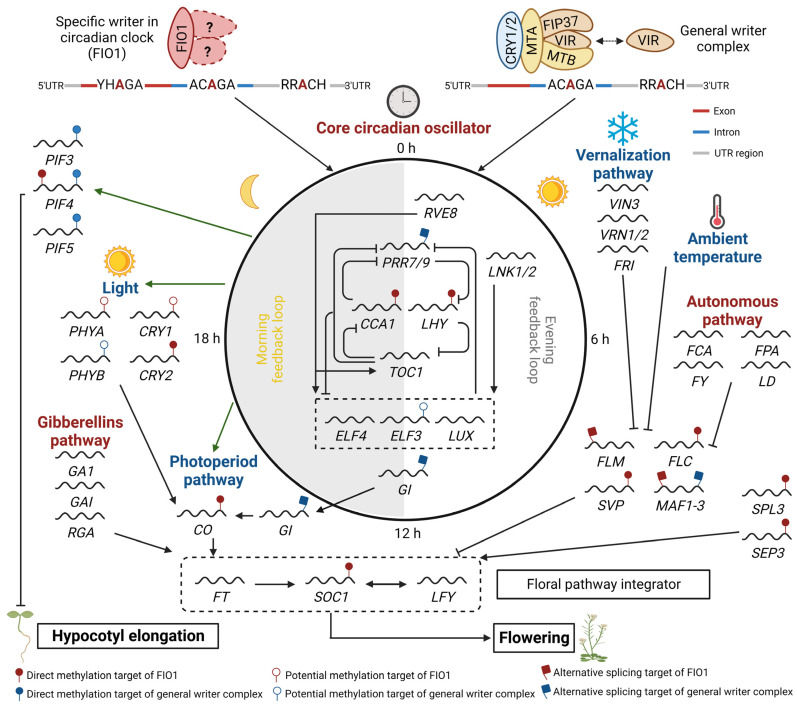
A proposed model depicting the involvement of FIO1 and general m^6^A writer complexes in regulating circadian clock, flowering, and hypocotyl elongation in Arabidopsis. Diverse pathways respond to various external (photoperiod, vernalization, and ambient temperature) and internal (autonomous, gibberellins, and circadian oscillator) stimuli/signals to regulate these processes are shown in red and blue, respectively. FIO1 prevents premature flowering and maintains the circadian period by affecting the expression level, splicing, and/or stability of several key circadian clock transcripts (e.g., *LHY* and *CCA1*), flowering regulator transcripts (e.g., *SOC1* and *FLC*), and photomorphogenesis-related genes (e.g., *PIF4*) through direct m^6^A methylation of the 3′ UTR and a subset on the CDS region, or indirectly through m^6^A methylation of U6 snRNA. The general m^6^A writer complex (including MTA, MTB, FIP37, VIR, and HAKAI) recognition motifs are mainly present in the 3′ UTR region. PIF37 can interact with cryptochromes (CRY1) and increases the m^6^A modification of *PIF3*, *PIF4*, and *PIF5*, which consequently reduces their RNA stability and promotes photomorphogenesis [111]. However, CRY2 recruits MTA, MTB, and PIF37 to methylate several core circadian clock genes and enhances their mRNA stability [62]. The involvement of other m^6^A writer components such as VIR and HAKAI in the regulation of these processes requires further investigation.

**Figure 3 plants-12-00624-f003:**
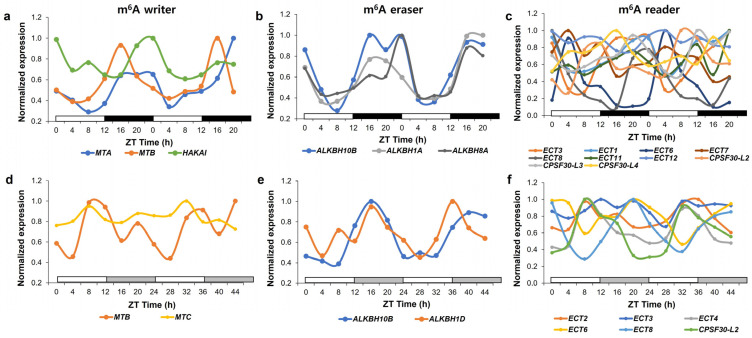
Expression of genes encoding proteins involved in m^6^A modification in plants under diurnal and free-running conditions. Relative expression of rhythmic genes in the m^6^A writer (**a**,**d**), m^6^A eraser (**b**,**e**), and m^6^A reader (**c**,**f**) in diurnal (**a**–**c**) and free-running conditions (**d**–**f**) was determined as a normalized value against the maximum value of expression at all-time points. Genes involved in m^6^A metabolism are selected according to [132]. Raw expression data were derived from the DIURNAL project (http://diurnal.mocklerlab.org, accessed on 2 January 2023) and rhythmic genes in diurnal and free-running conditions were determined based on the correction cutoff value = 0.8 and the condition = LDHC and LL23_LDHH (the DIURNAL project), respectively [131].

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
