# Peer review of "Current Insights into m6A RNA Methylation and Its Emerging Role in Plant Circadian Clock"

_plants, 2023, doi:10.3390/plants12030624_

Round 1

Reviewer 1 Report

In this study, Chuong et al summarized the recent insights into the roles of N6-adenosine methylation (m6A) in regulation of RNA, and discussed the potential function of m6A modification in regulating plant circadian clock. This review are well-written and two elegant schematic model for the regulatory roles of m6A in RNA metabolism and the potential mechanism of m6A regulators involved in circadian oscillation were provided. Additionally, this review provided an overview of the circadian response of m6A regulator genes (writers, readers and are erasers), which is interesting for further study of the m6A dynamic regulation during a round of 24-h circle. Overall, the review is well-organized and the reviewer recommend for publication of this review on the Plants Journal.

Some minor issues:

  1. In line 60-65, for the strategies/methods to identify or recognize m6A sites, please also include the literature of Nanopore-based direct RNA sequencing (DRS) for m6A site identification, which also have been applied to plant. 
  2. In Line 612, is “ALBHK” a typo of ALKBHs?
  3. In figure 1, CPSF30 should be CPSF30-L, which contains YTH domain.

Author Response

Response Letter to Reviewer’s Comments

“Current insights into m6A RNA methylation and its emerging role in plant circadian clock”

 # Reviewer 1

In this study, Chuong et al summarized the recent insights into the roles of N6-adenosine methylation (m6A) in regulation of RNA, and discussed the potential function of m6A modification in regulating plant circadian clock. This review are well-written and two elegant schematic model for the regulatory roles of m6A in RNA metabolism and the potential mechanism of m6A regulators involved in circadian oscillation were provided. Additionally, this review provided an overview of the circadian response of m6A regulator genes (writers, readers and are erasers), which is interesting for further study of the m6A dynamic regulation during a round of 24-h circle. Overall, the review is well-organized and the reviewer recommend for publication of this review on the Plants Journal.

Response: We would like to thank the reviewer for his/her positive evaluation and for the constructive comments and suggestions that have helped us improve the quality of the manuscript. We have revised the manuscript following your suggestions and comments to improve its quality. All the changes made in responses to the reviewer’s comments were highlighted in red in the file “m6A review_Plants_Revised_Highlighted Changes”. We hope that our revised version would now meet your expectation. Please see below our responses itemized to your comments and suggestions.

Comment 1: In line 60-65, for the strategies/methods to identify or recognize m6A sites, please also include the literature of Nanopore-based direct RNA sequencing (DRS) for m6A site identification, which also have been applied to plant.

Response: We would like to thank the reviewer for this suggestion. We have included the literature of Nanopore-based direct RNA sequencing for m6A site identification in our manuscript and reference list as followed:

 “Based on the strategies to identify or recognize m6A sites, these approaches can be broadly divided into two categories: antibody-dependent methods such as MeRIP-seq [18], miCLIP-seq [24], or SLIM-seq [25] and antibody-independent methods such as DART-seq [26], m6A-SEAL-seq [27], m6A-SAC-seq [28], or nanopore-based direct RNA-seq [29]. Each of these methods have their own advantages and limitations, which have been discussed in detail in a previous review [30,31].” (L60-64)

  • 29. Garalde, D.R.; Snell, E.A.; Jachimowicz, D.; Sipos, B.; Lloyd, J.H.; Bruce, M.; Pantic, N.; Admassu, T.; James, P.; Warland, A.; et al. Highly parallel direct RNA sequencing on an array of nanopores. Nat. Methods 2018, 15, 201-206, doi:10.1038/nmeth.4577.
  • 31. Zhao, X.; Zhang, Y.; Hang, D.; Meng, J.; Wei, Z. Detecting RNA modification using direct RNA sequencing: A systematic review. Comput. Struct. Biotechnol. J. 2022, 20, 5740-5749, doi:10.1016/j.csbj.2022.10.023.

Comment 2: In Line 612, is “ALBHK” a typo of ALKBHs?

Response: We would like to thank the reviewer for this inquiry. We have corrected this detail as below:

“Plant erasers contain only members from the ALBHKs family. m6A readers are responsible for recognizing m6A-containing transcripts.” (L612-614).

Comment 3: In figure 1, CPSF30 should be CPSF30-L, which contains YTH domain.

Response: We highly appreciate the reviewer for this important correction. We have corrected this detail in Figure 1.

Reviewer 2 Report

The authors of this review article have compiled and summarized recent 24 insights into the molecular mechanisms behind m6A modification and its various roles in the regulation of RNA.  They discussed the potential role of m6A modification in regulating the plant circadian clock. They further discussed the potential future directions for the study of mRNA modification in plants and how they can aid in the understanding of the plant circadian clock.

The article is well-written and all relevant and all recent developments and insights are appropriately compiled and summarized. I do not have any further comments.

Author Response

Response Letter to Reviewer’s Comments

“Current insights into m6A RNA methylation and its emerging role in plant circadian clock”

 # Reviewer 2

The authors of this review article have compiled and summarized recent 24 insights into the molecular mechanisms behind m6A modification and its various roles in the regulation of RNA.  They discussed the potential role of m6A modification in regulating the plant circadian clock. They further discussed the potential future directions for the study of mRNA modification in plants and how they can aid in the understanding of the plant circadian clock.

The article is well-written and all relevant and all recent developments and insights are appropriately compiled and summarized. I do not have any further comments.

Response: We thank the reviewer for the positive feedback.

Reviewer 3 Report

A review by Nguyen Nguyen Chuong et al “Current Insights into m6A RNA Methylation and Its Emerging Role in Plant Circadian Clock” considers various aspects of m6A methylation in plants and its relation to the functioning of the intrinsic circadian clock in plants. The review is quite interesting, covering the latest data in this area and collected information could be valuable for the research community. The manuscript is carefully and comprehensibly written, although some of the sentences seem unnecessarily cumbersome. The figures illustrate the given data quite well. The Conclusion reflects the meaning of the text and, most importantly, outlines future research and approaches.

I believe that this manuscript may be accepted for publication in MDPI Plants in its present form.

Little remarks:

Lines 216 and 219: ETC or ECT?

Lines 385-386 “…indicating an involvement of CRY1/2 in parametric entrainment of blue light input to the circadian clock” - The sentence is not clear, please rewrite.

Lines 403-404: VRIMA or VIRMA?

Author Response

Response Letter to Reviewer’s Comments

“Current insights into m6A RNA methylation and its emerging role in plant circadian clock”

 # Reviewer 3

A review by Nguyen Nguyen Chuong et al “Current Insights into m6A RNA Methylation and Its Emerging Role in Plant Circadian Clock” considers various aspects of m6A methylation in plants and its relation to the functioning of the intrinsic circadian clock in plants. The review is quite interesting, covering the latest data in this area and collected information could be valuable for the research community. The manuscript is carefully and comprehensibly written, although some of the sentences seem unnecessarily cumbersome. The figures illustrate the given data quite well. The Conclusion reflects the meaning of the text and, most importantly, outlines future research and approaches.

I believe that this manuscript may be accepted for publication in MDPI Plants in its present form.

Response: We would like to thank the reviewer for his/her positive evaluation and for the constructive comments and suggestions that have helped us improve the quality of the manuscript. We have revised the manuscript following your suggestions and comments to improve its quality. All the changes made in responses to the reviewer’s comments were highlighted in red in the file “m6A review_Plants_Revised_Highlighted Changes”. We hope that our revised version would now meet your expectation. Please see below our responses itemized to your comments and suggestions.

Comment 1: Lines 216 and 219: ETC or ECT?

Response: We would like to thank the reviewer for this correction. We have corrected this detail in our manuscript as followed:

 “The conserve consensus sequence RRm6ACH is recognized by the YTH domain of ECTs. Concurrently, the intrinsic disorder region within the ECTs forms a stable interaction with the U-rich sequence in adjacent regions, thereby controlling the occupancy of the binding adjacent to m6A with different RNA-binding factors.” (L215-218)

Comment 2: Lines 385-386 “…indicating an involvement of CRY1/2 in parametric entrainment of blue light input to the circadian clock” - The sentence is not clear, please rewrite.

Response: We would like to thank the reviewer for this constructive suggestion. Followed your advice, we have revised the sentence as below:

 “The cry1cry2 double mutant showed reduced responses to period shortening effects by increased intensity of blue light, indicating an involvement of CRY1 and CRY2 in parametric entrainment of blue light input to the circadian clock through continuous modulation of the circadian clock by recognizing intensity of blue light.(L384-387)

Comment 3: Lines 403-404: VRIMA or VIRMA?

Response: We would like to thank the reviewer for this correction. We have corrected this detail in our manuscript as followed:

 “Additionally, vir-1 mutation abolished the 3′ end formation in mRNAs, which is similar to the function of human VIRMA, an orthologue of VIR, in the polyadenylation selection [131].” (L403-405)
